Expression patterns of E2Fs identify tumor microenvironment features in human gastric cancer

Li Fanni 1
Yan Jun 2
Leng Jing 2
Yu Tianyu 2
Zhou Huayou 3
Liu Chang 2
Huang Wenbo 2
Sun Qi sunqi4875@163.com 2
Zhao Wei zhaowei9803@126.com 2
1 Department of Talent Highland, The First Affiliated Hospital of Xi’an Jiaotong University , Xi’an , China
2 Department of General Surgery, The First Affiliated Hospital of Xi’an Jiaotong University , Xi’an , China
3 Department of General Surgery, Hanzhong Central Hospital , Hanzhong , China
Banerjee Priyanka
Electronic publication date: 2024 Feb 13
Publication date: 2024
Volume: 12
Electronic Location ID: e16911
Received 2023 Jul 28; Accepted 2024 Jan 17
Copyright: ©2024 Li et al.
Copyright year: 2024
Copyright holder: Li et al.
License: This is an open access article distributed under the terms of the Creative Commons Attribution License, which permits unrestricted use, distribution, reproduction and adaptation in any medium and for any purpose provided that it is properly attributed. For attribution, the original author(s), title, publication source (PeerJ) and either DOI or URL of the article must be cited.
License URL: https://creativecommons.org/licenses/by/4.0/

Keywords: E2F, Gastric cancer, Tumor microenvironment, Stroma, Immunotherapy

Funding: National Natural Science Foundation of China 82073271 81803026 81702362 Science and Technology Program of Shaanxi Province 2023KJXX-031 Key Research and Development Program of Shaanxi 2019SF-057 Institutional Foundation of The First Affiliated Hospital of Xi’an Jiaotong University YXJLRH2022044 2022YQPY07 2022MS-18 Shaanxi Provincial Innovative Talents Promotion Plan 2022TD-58 This work was supported by the National Natural Science Foundation of China (82073271, 81803026, 81702362), the Science and Technology Program of Shaanxi Province (2023KJXX-031), the Key Research and Development Program of Shaanxi (No. 2019SF-057), the Institutional Foundation of The First Affiliated Hospital of Xi’an Jiaotong University (YXJLRH2022044, 2022YQPY07, 2022MS-18), and the Shaanxi Provincial Innovative Talents Promotion Plan (2022TD-58). The funders had no role in study design, data collection and analysis, decision to publish, or preparation of the manuscript.

==============================
Objective

E2F transcription factors are associated with tumor development, but their underlying mechanisms in gastric cancer (GC) remain unclear. This study explored whether E2Fs determine the prognosis or immune and therapy responses of GC patients.

Methods

E2F regulation patterns from The Cancer Genome Atlas (TCGA) were systematically investigated and E2F patterns were correlated with the characteristics of cellular infiltration in the tumor microenvironment (TME). A principal component analysis was used to construct an E2F scoring model based on prognosis-related differential genes to quantify the E2F regulation of a single tumor. This scoring model was then tested in patient cohorts to predict effects of immunotherapy.

Results

Based on the expression profiles of E2F transcription factors in GC, two different regulatory patterns of E2F were identified. TME and survival differences emerged between the two clusters. Lower survival rates in the Cluster2 group were attributed to limited immune function due to stromal activation. The E2F scoring model was then constructed based on the E2F-related prognostic genes. Evidence supported the E2F score as an independent and effective prognostic factor and predictor of immunotherapy response. A gene-set analysis correlated E2F score with the characteristics of immune cell infiltration within the TME. The immunotherapy cohort database showed that patients with a higher E2F score demonstrated better survival and immune responses.

Conclusions

This study found that differences in GC prognosis might be related to the E2F patterns in the TME. The E2F scoring system developed in this study has practical value as a predictor of survival and treatment response in GC patients.

Introduction

Gastric cancer (GC) is the second leading cause of cancer death globally, with one million new cases diagnosed and 700,000 deaths annually (Bray et al., 2018). There are multiple treatment methods for gastric cancer, such as chemotherapy, radiotherapy, immunotherapy, and targeted therapy (Joshi & Badgwell, 2021), and immune checkpoint inhibitors have introduced new possibilities for cancer treatment (Abdel-Rahman, 2016). However, some molecular characteristics of GC remain uncertain that could contribute to the discovery of innovative pathogenetic and therapeutic targets, improving treatment efficacy and patient prognosis and survival (Chen et al., 2022). Given the heterogeneity of GC patients, models that accurately predict treatment effects are also urgently needed to guide clinical strategies.

The E2F family of transcription factors consists of eight pivotal members (E2F1, E2F2, E2F3, E2F4, E2F5, E2F6, E2F7, E2F8) that possess highly similar DNA binding domains with an ability to bind target promoters and regulate their expression (Chen, Tsai & Leone, 2009; Kent & Leone, 2019; Logan et al., 2005). E2Fs play a critical role in cell cycle control through multiple mechanisms, such as the cyclin-dependent kinase (CDK)–RB–E2F axis, whose misexpression inevitably provokes an oncogenic effect (Di Fiore et al., 2013). In addition to the cell cycle, E2F transcription factors also are also involved in biological processes on the pathways of malignant progression (e.g., apoptosis, autophagy, angiogenesis, and metabolism) (Lammens et al., 2009; Schaal, Pillai & Chellappan, 2014; Phillips & Vousden, 2001). There is growing evidence that high expression levels of E2F family members may promote the occurrence and development of some tumors. It has been reported that overexpression of E2Fs is associated with poor overall survival (OS) in patients with solid tumors (Lan et al., 2018; Manicum et al., 2018; Yao, Lu & Shao, 2020). E2F1 is the most extensively studied E2F family member in GC, which is frequently overexpressed in human gastric adenocarcinoma. The cancer-promoting mechanisms of E2F1 are thought to increase P-glycoprotein (P-gp) levels. The upregulation of P-gp decreases intracellular accumulation of chemotherapeutic drugs and contributes to the survival of tumor cells (Yan et al., 2014).

Although the E2Fs have been reported as cancer promoters in various tumors, the biological function of these transcription factors in GC and their roles in forming the tumor microenvironment (TME) is not entirely understood (Kitajima, Li & Takahashi, 2020; Woo et al., 2000). Based on data from several large public databases, this study systematically investigated the regulation patterns of eight E2F transcription factors in GC and further explored their correlation with prognosis, TME, and treatment response.

Materials and Methods

Data acquisition of gastric cancer samples

Training data sets were retrieved using the R package “TCGAbiolinks” from The Cancer Genome Atlas Stomach Adenocarcinoma (TCGA-STAD) database, including transcriptome data,Fragments Per Kilobase of exon model per Million mapped fragments (FPKM), and single nucleotide variations (SNVs). The official corrected survival information (overall survival, OS) and clinical information (age, stage, gender, grade) were downloaded from the TCGA Pan-Cancer Clinical Data Resource (TCGA-CDR). A total of 351 tumor samples with both expression and survival information were retained for subsequent analysis.

The validation data sets, comprising six independent cohorts (GSE13861, GSE15459, GSE26899, GSE26901, GSE66229, GSE84433) were downloaded from the Gene Expression Omnibus (GEO).

The validation cohort for immunotherapy was acquired using the R package “IMvigor210CoreBiologies,” which included transcriptomic data and clinical information of 298 urothelial carcinoma patients who underwent anti-PD-1/PD-L1 therapy.

Unsupervised clustering of transcription factors and differentially expressed genes

The R package “ConsensusClusterPlus,” along with the unsupervised clustering method, was used to divide patients into distinct E2F clusters based on E2F expression levels, which was then repeated 1,000 times to verify the stability of the classification. Gene cluster count was determined using the same methods asE2F.

Identification of the immune characteristics of clusters

Single-sample gene-set enrichment analysis (ssGSEA) was employed to assess the relative abundance of immune cell infiltration in TME, which comprised 28 human immune cell types, including activated CD8 T cells, activated dendritic cells, and macrophages (Charoentong et al., 2017).

The ESTIMATE algorithm was adopted to calculate the immune score and stromal score of each tumor sample. The Wilcoxon test was then used to compare the differences in the scores between groups.

Selection and identification of differentially expressed genes between distinct subgroups

The R package “limma” and empirical Bayesian algorithm were adopted to filter out differentially expressed genes (DEG) between diverse E2Fclusters, which were defined as E2F-related genes, using the following screening criteria: fold change value ≥ 1.5 (—log2FC—>=0.585) and a significance threshold (FDR<0.05). These genes were further collected through a univariate Cox regression analysis to obtain a refined prognosis-related gene list. Then, the samples were divided into various gene clusters by applying an unsupervised clustering method according to the expression levels of genes ultimately screened out.

Construction of E2F scoring system

A principal component analysis (PCA) was employed to construct an E2F scoring system according to E2F prognostic-related genes through a dimension reduction method to transform many indicators into a couple of comprehensive indicators. Both principal components 1 and 2 were selected to act as E2F scores (Sotiriou et al., 2006). The advantage of this approach is that it retains the critical genetic characteristics while removing non-essential genetic information. The computed formula was as follows: E2F score= ∑(PC1i + PC2i)

where i represents the E2F prognostic-related gene expression level.

Analysis of functional mechanisms

Gene set variation analysis (GSVA) is a non-parametric unsupervised strategy to estimate changes in the activity of pathways and biological processes in samples, and was used in this study to determine underlying gene sets with E2F score changes. The hallmark gene set (msigdb.v7.4.symbols.gmt) was downloaded from the Molecular Signatures Database (MSigDB) for GSVA. Statistical significance was considered at P ≤ 0.05.

A Gene Ontology (GO) enrichment analysis involving molecular functions (MFs), biological pathways (BPs), and cellular components (CCs) and a Kyoto Encyclopedia of Genes and Genomes (KEGG) enrichment analysis centering gene function were used to annotate the DEGs of distinct E2F phenotypes with p-value cutoff = 0.05 and pAdjustMethod = “BH” using the R package “clusterProfiler.”

Tumor drug sensitivity and efficacy prediction

Based on the Genomics of Drug Sensitivity in Cancer (GDSC) dataset, the sensitivity of chemotherapy drugs was measured by calculating the IC50 in the training set using the R package “oncoPredict” and the “calcPhenotype” algorithm.

The immunotherapy validation cohort was obtained using the R package “IMvigor210CoreBiologies,” which included transcriptomic data and clinical information of 298 urothelial carcinoma patients who underwent anti-PD-1/PD-L1 therapy.

Quantitative real-time PCR

RNA was extracted from both normal and tumor tissues of 12 GC patients randomly selected from the tissue bank of the First Affiliated Hospital of Xi’an Jiaotong University. Written, informed consent was obtained from all patients. Ethical approval for the study was obtained from the Ethics Committee of The First Affiliated Hospital of Xi’an Jiaotong University (No. XJTU1AF2023LSK-451). The clinical information of the participating patients are listed in Table S1. TRIzol reagent was used to extract total RNA from normal tissues and tumor tissues, and spectrophotometry was used for quantification and quality control. Quantitative PCR analysis was performed with the SYBR-Green PCR master mix (TransGen, Beijing, China) on a CFX96 real-time PCR detection system (BioRad, United States). The primer sequences of E2F2, E2F8, and GAPDH are listed in Table S2. Fold differences of mRNA were calculated using the 2−ΔΔCT method and normalized to GAPDH levels. The same methods were used to determine the mRNA level of genes E2F2 and E2F8 in cells.

Cell phenotype assay

Human stomach carcinoma cell line AGS was cultured in RPMI 1640 medium (Gibco, United States) with 10% fetal bovine serum (FBS, Sijiqing, Hangzhou, China) for in vitro experiments. AGS cells were then seeded into corresponding microplates for further detection 24 h after transfecting siRNA with RNATransMate (Sangon, Shanghai, China). Cell proliferation assays were performed using a Cell Counting Kit-8 (CCK8) colorimetric assay (Fude, Hangzhou, China) at a wavelength of 450 nm by a Spark microplate reader (TECAN, Switzerland). A colony formation experiment was also performed to determine cell proliferation. For transwell migration assays, transfected cells were seeded into 8-µm pore inserts (Falcon, United States). To induce cell migration, 1% FBS was added to the top chamber, and 10% FBS was added to the bottom chamber. After 48 h of migration, the cells were stained with 0.1% crystal violet (Biosharp, Anhui, China) and observed under a microscope. Cell counts were obtained using ImageJ software. The data was evaluated by normalizing to cells transfected with the negative control siRNA. The siRNA sequences of E2F2 and E2F8 are listed in Table S3. The overexpressing plasmids were purchased from Weizhen Biosciences (Jinan, China) and then used to transfect AGS cells to construct cell lines overexpressing E2F2 and E2F8. All plasmids were confirmed by DNA sequencing, and detailed plasmid sequence information is included in Table S4. After transfection by Lipo3000 (Invitrogen, Waltham, MA, USA), cells were used for proliferation and migration assays. The experiment and analysis methods are the same as above.

Statistical analysis

All statistical analyses were performed by the R software package, version 4.1.2 (R Core Team, 2021). Nonparametric tests, such as the Wilcoxon rank-sum test (comparison between two groups) or the Kruskal–Wallis test (comparison among multiple groups), were used for continuous variables (such as expression quantity and infiltration ratio). The survival analysis was performed using a Kaplan–Meier (K-M) procedure (log-rank test). The univariate Cox regression model was employed to calculate hazard ratios, while the multivariable Cox regression model was adopted to identify independent prognostic factors. All statistical tests were two-sided, with statistical significance set at P ≤ 0.05.

Results

Expression profile and heritable variation of E2Fs in gastric cancer

This study identified the significant differential expression of total E2F transcription factors (E2F1–E2F8) between GC and normal samples from the TCGA dataset, suggesting the dysregulation of E2F expression plays a crucial role in the occurrence and development of GC (Fig. 1A).

Figure 1 Genetic variation of E2F transcription factors in gastric cancer.

(A) Differential expressions of E2Fs in normal and tumor tissues. (B) Delta beta values of methylation signals in the promoter region. (C) Somatic mutation frequency. (D) Heatmap of transcription factor expression correlation. ∗P < 0.05, ∗∗P < 0.01, ∗∗∗P < 0.001, ∗∗∗∗P < 0.0001, ns means “not statistically significant”.

A methylation analysis of these normal and tumor samples showed that the probes in the 1,500 bp range of the eight transcription factor promoter regions had minor changes. Even the highest E2F6 had a change of only 0.2 (Fig. 1B). The somatic mutation frequency of eight E2Fs in GC samples from the TCGA dataset were then calculated. Somatic mutation was observed at a low frequency. Only E2F7 reached 2.17%, and other E2Fs reached almost zero (Fig. 1C). Spearman correlation was calculated for the expression level of E2Fs, and there was a significantly positive correlation between the eight factors, indicating potential synergy (Fig. 1D). These results suggest a limited contribution of the E2F transcription factor family to the tumor immune microenvironment (TIME) at the genomic level, indicating the influence of E2Fs on TIME may mainly be reflected in the expression level.

Association of E2F with immunity and prognosis

Based on The Cancer Immunome Atlas, an ssGSEA analysis was used to profile the infiltration degree of 28 kinds of immune cells in GC samples, most of which were then verified to be correlated with the expression level of E2Fs (Fig. 2A).

Figure 2 Correlation of E2F transcription factors with immune cells and patient prognosis.

(A) Heatmap of E2Fs and immune cells. (B) Prognostic forest map of E2Fs. ∗P < 0.05, ∗∗P < 0.01, ∗∗∗P < 0.001, ∗∗∗∗P < 0.0001, ns means “not statistically significant”.

The samples were then divided into higher- and lower-expression groups according to the median expression level of each transcription factor and the clinical and transcriptome data of the TCGA-STAD cohorts were transformed into K-M survival curves to estimate the prognostic value of the eight E2Fs (Fig. S1). Univariate Cox regression demonstrated that E2F2 and E2F8 showed a protective influence on survival, but the other E2Fs showed no independent contribution to survival (Fig. 2B). These results indicate that the eight E2Fs played a role in the regulation of TIME through a collaborative model and further affected clinical outcomes.

Identification of consensus clusters of E2F and corresponding clinical features

An unsupervised consensus clustering analysis identified two subgroups (Cluster1 and Cluster2) of GC patients associated with the expression level of the eight E2Fs supported by the cumulative distribution functions (CDF; Fig. 3A and Fig. S2). PCA was also conducted to verify the independence and diversity of the transcriptional profile in the two clusters (Fig. 3B). The differential expression levels of the eight E2Fs between the two E2FClusters are shown in the heat map in Fig. 3C. Poor tumor differentiation (Grade 3/4) dominated Cluster2, and patients with GC molecular subtype were characterized by Cluster2’s expression pattern, while patients with HM-indel and HM-SNV subtypes aligned with Cluster1’s expression pattern. Previous studies regarding molecular subtypes suggest that GC subtypes are significantly associated with adverse clinical outcomes, whereas MSI (HM-indel and HM-SNV) contribute to improved survival (Sohn et al., 2017; Cristescu et al., 2015).

Figure 3 Classification of E2F clusters and clinical characteristics of gastric cancer patients.

(A) Consensus clustering matrix for k = 2. (B) Principal component analysis of the sample distribution of the E2F clusters. (C) The heatmap showed the expression level of eight E2F transcription factors in distinct subtypes. Patient cluster, age, sex, grade, molecular subtype, and histological type were included. Red/blue represents high/low expression of E2Fs. (D) Kaplan–Meier analysis of overall survival in E2F clusters of patients with gastric cancer. (E) Multivariate Cox regression analysis of prognosis factors. ∗P < 0.05, ∗∗P < 0.01, ∗∗∗P < 0.001, ∗∗∗∗P < 0.0001, ns means “not statistically significant”.

K-M curves confirmed that between the two types of E2FClusters identified, Cluster1, with 229 samples, had better survival than Cluster2, with 122 samples (Fig. 3D). A multivariate Cox analysis combining multiple clinical features revealed that the expression patterns of the transcription factors identified were independent prognostic factors (Fig. 3E). These results indicate that the Cluster2 E2F expression pattern is an independent risk factor for GC prognosis.

Immune infiltration and genetic differences in E2F clusters

Cluster2 had elevated immune scores along with stromal scores using the “ESTIMATE” algorithm to assess immune cell infiltration (Fig. 4A). Previous studies have found that immune cells can infiltrate tumors with immune rejection phenotype, but they are confined to the fibroblast-and collagen-rich peritumoral stroma and cannot penetrate into the core of tumor cells (Mariathasan et al., 2018). Activated stromal cells in the TME are known to inhibit T cells (Lakins et al., 2018), so stromal activation in Cluster2 may inhibit the anti-tumor effect of immune cells. A hallmark pathway analysis demonstrated that stromal activity in Cluster2 was significantly enhanced, including epithelial-mesenchymal transition (EMT), transforming growth factor-β (TGF-β), angiogenesis, and many more, confirming that stromal activation may help explain the negative outcomes of patients in Cluster2 (Fig. 4B).

Figure 4 The estimation of tumor microenvironment and genetic differences between E2F subtypes.

(A) The ImmuneScore and StromalScore of the two E2Fclusters. (B) Heat map of hallmark pathway activity between E2F subtypes. (C) Heat map of differentially expressed genes between E2F subtypes. (D) Bubbles of KEGG enrichment analysis for differentially expressed genes. (E) Forest map of prognostic efficacy of the top 20 prognostic genes. (F) Correlation heat map of E2Fs and gene clusters. (G) Kaplan–Meier analysis of overall survival in gene clusters of patients with gastric cancer. ∗P < 0.05, ∗∗P < 0.01, ∗∗∗P < 0.001, ∗∗∗∗P < 0.0001, ns means “not statistically significant”.

To further investigate the potential biological roles of genes characterized by E2Fs mode, a total of 1,340 differentially expressed genes (DEGs) were screened between the two clusters (Fig. 4C). A KEGG analysis indicated that these genes are involved in the cell cycle, DNA replication, mismatch repair, and other cell progressions, confirming that E2F transcription factors are involved in the immune regulation of TME (Fig. 4D). According to the unsupervised clustering of these E2F DEGs, the 351 GC patients were classified into three E2F-related gene clusters (E2F gene clusters 1, 2, and 3), which were strongly correlated with the expression of the eight E2Fs. The K-M curves indicated that these gene clusters were also correlated with survival status: E2F gene cluster 1 had the best OS when compared to clusters 2 and 3 (Figs. 4F–4G and Fig. S3). A univariate Cox analysis was performed on the DEGs and 58 were found to be correlated with prognosis (Fig. 4E).

Establishment and validation of the E2F scoring system

PCA was used to calculate an E2F score based on the 58 genes associated with prognosis to quantify the E2F transcription factor expression patterns in individual gastric tumors. Cluster1 scored significantly higher than Cluster2 (Fig. 5A). Also, E2F gene cluster 1 scored the highest among the three gene subtypes. Based on the median PCA score, these 351 patients were stratified into two groups (high and low E2F score). The OS of the low-E2F score group was worse than the high-E2F score group (Fig. 5B). A multivariate Cox analysis showed that E2F score was an independent prognostic factor, indicating that the E2F score model could characterize the prognosis of the sample with decent accuracy (Fig. 5C). These associations were then validated in six independent GEO datasets (GSE13861, GSE15459, GSE26899, GSE26901, GSE66229, and GSE84433). All the validation sets revealed consistent findings: patients with a high E2F score had a distinctly better prognosis than patients with a low E2F score, which reflected the stability and credibility of the E2F scoring system (Fig. 5D).

Figure 5 Construction and verification of the E2F scoring system.

(A) The E2F scores in distinct E2F Clusters and gene clusters. (B) Kaplan–Meier analysis of overall survival in E2F scoring system of patients with gastric cancer. (C) Multivariate Cox regression analysis of prognosis factors for the E2F scoring system. (D) Kaplan–Meier analyses of the E2F scoring system in six validation cohorts. ∗P < 0.05, ∗∗P < 0.01, ∗∗∗P < 0.001, ∗∗∗∗P < 0.0001, ns means “not statistically significant”.

Molecular mechanism of the E2F scoring system

A gene-set enrichment analysis was used to assess molecular pathway enrichment to help explain the mechanisms behind the E2F score. The pathway analysis results showed that the group of genes with low E2F scores were enriched in matrix activation pathways involving the calcium signaling pathway, adhesion molecules, extracellular matrix (ECM) receptor interaction, and the MAPK signaling pathway (Fig. 6A). A correlation analysis between E2F score and hallmark pathway analysis indicated that E2F score was negatively correlated with the matrix activation pathways, such as the NOTCH pathway and IL2 STAT5 pathway, and positively correlated with immune-related pathways, such as DNA repair (Fig. 6B). In terms of immune cell infiltration, levels of activated CD4 T cell infiltration in the high-E2F score group were significantly higher than in the low-E2F score group, while the levels of T helper cell infiltration and other immune cell infiltration were significantly lower in the high-E2F score group than in the low-E2F score group (Figs. 6C–6D). Additionally, a higher tumor mutational burden (TMB) was observed in the high-E2F score group (Fig. 6E). TMB leads to tumor-specific and potentially highly immunogenic neoantigens, which can be recognized by the major histocompatibility complex (MHC) to enhance the immune response and improve prognosis. The distribution of these different classification systems and subtypes are shown in the alluvial diagram in Fig. 6F.

Figure 6 Molecular mechanism of the E2F scoring system.

(A) GSEA enrichment of high- and low-E2F score groups. (B) Correlation between hallmark pathway activity and E2F score. (C) Comparison of innate immune cell infiltration in different E2F score groups. (D) Comparison of adaptive immune cell infiltration in different E2F score groups. (E) TMB of high- and low-E2F score groups. (F) The alluvial diagram of E2F expression in groups with distinct E2Fclusters, gene clusters, E2F score, molecular subtype, and survival status. ∗P < 0.05, ∗∗P < 0.01, ∗∗∗P < 0.001, ∗∗∗∗P < 0.0001, ns means “not statistically significant”.

Treatment response prediction and validation of the E2F scoring system

The E2F scoring system reflected the regulation and modulation of the TIME, so this study further explored the promising role of E2F score in predicting therapeutic response and monitoring the effectiveness of treatments. Combined with drug information and expression profiles in the training set samples of GDSC, the Spearman correlation between the IC50 of several common GC drugs and E2F score was calculated. The results showed that the IC50 values of 5-Fluorouracil, Cisplatin, Docetaxel, and Paclitaxel were significantly negatively correlated with E2F score (Fig. 7A). Additionally, in the immunotherapy cohort database, a high E2F score was correlated with a better OS when compared with a low E2F score (Fig. 7B). When assessing patient response to immunotherapy, responders had significantly higher average E2F scores than non-responders (Fig. 7C). There were also significant differences in the distribution of high and low E2F scores among patients with distinct responses, and most non-responders had low E2F scores (Fig. 7D). These results showed that the higher the E2F score, the higher the sensitivity to the drugs, and the better the patient prognosis, confirming that E2F score is an underlying indicator for predicting chemotherapy and immunotherapy responses.

Figure 7 Efficacy and validation of the E2F scoring system for treatment response prediction.

(A) Scatter plot of correlation between chemotherapeutic drugs log2(IC50) and E2F score. (B) Kaplan–Meier analysis of E2F score in immunotherapy cohort. (C–D) E2F score distribution for different treatment response groups. ∗P < 0.05, ∗∗P < 0.01, ∗∗∗P < 0.001, ∗∗∗∗P < 0.0001, ns means “not statistically significant”.

In vitro validation

Given the independent prognostic role of E2F2 and E2F8 in the database, these two transcription factors were chosen for validation in the collected samples. The qPCR assay results showed the mRNA levels of E2F2 and E2F8 were significantly upregulated in the gastric cancer tissues compared with the corresponding normal tissues, which was consistent with both the transcriptomic database and the findings of previous studies (Fig. 8A).

Figure 8 In vitro validation of expression and function for transcription factor.

(A) The qPCR for E2F2 and E2F8 in normal tissues and GC tissues (n = 12, NT: normal tissues, T: tumor tissues). (B) The qPCR for E2F2 and E2F8 in AGS cell lines after transfection. (C) Cell proliferation assay of AGS cell lines after transfection. (D) Colony-formation assay of AGS cell lines after transfection. (E) Transwell migration of AGS cell lines after transfection, scale bar = 100 µm. (F) The qPCR for E2F2 and E2F8 in AGS cell lines after overexpression. (G) Cell proliferation assay of AGS cell lines after overexpression. (H) Colony-formation assay of AGS cell lines after overexpression. (I) Transwell migration of AGS cell lines after overexpression, scale bar =100 µm. (n = 6, NC: negative control, OE: overexpression) ∗P < 0.05 vs. NC, ∗∗P < 0.01 vs. NC, ∗∗∗P < 0.001 vs. NC, ns means “not statistically significant”.

To validate the prognostic implications of E2F2 and E2F8 in gastric cancer, E2F2 and E2F8 were transiently knocked down with siRNA in AGS cell lines, and the cells were phenotypically and functionally characterized. Transfection effectively suppressed the RNA levels of E2F2 and E2F8 compared to the siRNA negative control (Fig. 8B). Cell proliferation was assessed at 24 h, 48 h, and 72 h after transfection. Compared with the negative control, the knockdown of E2F2 resulted in the significant loss of tumor cell proliferation, while the knockdown of E2F8 increased tumor cell proliferation (Fig. 8C). Colony formation experiment was also performed to determine cell viability. E2F2-deficient cells formed significantly fewer colonies, but the opposite was observed in E2F8-deficient cells (Fig. 8D). The in vitro transwell migration assay results showed that the low expression of E2F2 inhibited migration, and the low expression of E2F8 induced migration (Fig. 8E). We further overexpressed E2F2 and E2F8 by transfecting AGS cells with corresponding plasmids, using the empty vector as control. The efficacy of overexpressed plasmids was validated by qPCR and the RNA levels of E2F2 and E2F8 significantly increased (Fig. 8F). Compared with the negative control, the overexpression of E2F2 accelerated tumor cell proliferation, while the overexpression of E2F8 inhibited tumor cell proliferation (Fig. 8G). As for Colony formation experiment, AGS cells overexpressing E2F2 displayed stronger cell viability than negative control, while overexpression of E2F8 reduced cell viability (Fig. 8H). In vitro cell migration experiments demonstrated that overexpressed E2F2 increased cell migration, and overexpressed E2F8 suppressed migration (Fig. 8I). These results suggested that E2F2 and E2F8 affect tumor cell migration and proliferation, supporting their prognostic impact.

Discussion

Previous research has demonstrated that the E2F family of transcription factors, a critical regulator of several cellular processes involving cell cycle control, apoptosis, DNA damage response, and many more, significantly impacts the pathogenesis and development of various tumors (Xu et al., 2021; Sun et al., 2019; Tsantoulis & Gorgoulis, 2005). One study found that E2F can serve as a prognostic biomarker and may become a potential therapeutic target for GC (Li et al., 2021b). There is also general agreement that both genetic alterations and immune infiltration in TME dominate all stages of tumor progression (Quail & Joyce, 2013). Although robust evidence has confirmed that the dysregulation of E2F expression promotes GC development and impacts treatment efficacy, the biological functions of the E2F family of transcription factors and their association with TME in GC remain ill-defined (Lee et al., 2008; Xanthoulis & Tiniakos, 2013). Identifying the roles of distinct E2F expression patterns in TME immune infiltration would aid in an understanding of E2F regulation and TME immune response, directing more effective treatment strategies.

Eight members of the E2F family have been identified and characterized (De Gregori & Johnson, 2006). Although some research indicates that dysregulation of E2F contributes to cancer, the current research is limited in scope (Dyson, 2016). Studies on GC, specifically, have centered on E2F1, and research on the other members of the E2F family is lacking (Fu et al., 2021; Lin et al., 2021). Based on the expression profiles of the eight E2Fs, this study revealed two independent patterns in GC, which differ in prognosis. Cluster2 showed higher stromal and immune scores and poorer survival rates. This phenomenon is explained by the immune-excluded phenotype, characterized by the presence of abundant immune cells and stromal cells (Chen & Mellman, 2017). Though there are large amounts of immune cells surrounding the tumor, they are trapped in the stroma of tumor cells rather than penetrating the parenchyma (Hegde, Karanikas & Evers, 2016). Given the deficiency of immune infiltration, T cells might be activated and proliferate when treated with anti-PD-L1/PD-1 agents, but with no immune response (Joyce & Fearon, 2015). Consistent with stromal activation, a hallmark analysis demonstrated that the stroma-related pathway was significantly enriched in Cluster2, involving the EMT, TGF-β, and angiogenesis pathways. These results suggest that the stromal activation in the TME may suppress anti-tumor immunity by limiting T cell infiltration (Mariathasan et al., 2018; Wang et al., 2018; Feng et al., 2018). These mechanisms help explain the differences in prognosis in patients in Cluster2 and validate the reliability of the clustering classification system.

Given that these classification systems were based on the patient population and therefore cannot precisely predict the E2F expression of an individual patient, an E2F score model was established using E2F-related prognostic genes, which could quantify E2F expression and eliminate individual heterogeneity. Among the included genes, CHAF1A was found to activate helicase activity of E2F transcription factors and participate in DNA repair (Mjelle et al., 2015). PDK4 level is regulated by E2F1 and influences tumor metabolism (Hsieh et al., 2008). Thus, the screened genes were verified to be associated with E2F regulation and tumor response. Cluster2 had a lower E2F score, which indicated a poorer prognosis. Gene cluster 2 and gene cluster 3 also had lower E2F scores and poorer prognoses. These results indicated the grouping methods established in this study were consistent with survival status. Multivariate analysis and multiple validation cohorts validated the E2F score as an independent prognostic factor. The GSEA demonstrated that the low E2F score group centered on the calcium signaling pathway, adhesion molecules, ECM receptor interaction, and MAPK signaling pathway, while the hallmark analysis showed that E2F score was inversely associated with the activation of the EMT, TGF-β, angiogenesis, NOTCH, and IL2 STAT5 pathways. Calcium signaling is a crucial messenger for cancer-associated fibroblasts (CAFs) to shape TME and promote the tumor process (Sadras, Monteith & Roberts-Thomson, 2021). Both cell adhesion molecules and angiogenesis are vital ingredients of the pro-tumorigenic extracellular matrix (Francavilla, Maddaluno & Cavallaro, 2009). Notch is a crucial regulator between the tumor and stroma, whose abnormal activation promotes tumor invasion and metastasis through angiogenesis, extracellular matrix remodeling, and immune suppression (Chimento et al., 2022; D’Assoro et al., 2022). There is robust evidence that the IL2 STAT5 pathway dominates the oncogenic functions of intracellular Matrix Gla protein, which also dynamically regulates effector and regulatory T cells (Wang et al., 2020; Jones, Read & Oestreich, 2020). These pathways enriched in the low E2F score groups participate in matrix activation and the immunosuppression of TME, so the low-E2F score group, corresponding with Cluster2, could be recognized as having an immune-excluded phenotype. In the IMvigor210 cohort, patients with a high E2F score receiving anti-PD-L1/PD-1 therapy showed better survival. Compared with other immune profiles, patients with inflamed tumors have a greater probability of clinical responses to immunotherapy (Tumeh et al., 2014; Taube et al., 2014). In this study, there were more patients that showed no response to immunotherapy in the low-E2F score group. Combined with the enriched pathway and clinical outcomes, the-low E2F score group could be recognized as having an immune-inflamed phenotype. The infiltration ratio of activated CD4 T cells was enhanced in the high-E2F score group, while the infiltration ratio of activated T regulatory cells was reduced. It is well known that CD4+T cells can mediate the destruction of tumor cells by recognizing antigen presentation, activating CD8+T cells, and enhancing cytokine activity (Joyce & Fearon, 2015; Saha, Martuza & Rabkin, 2017; Wong, Bos & Sherman, 2008). Conversely, regulatory T cells have a series of mechanisms through which they can inhibit cancer immune surveillance and trigger the immune escape of tumor cells (Agle et al., 2018; van der Vliet et al., 2007).

TMB was revealed to be positively associated with E2F score in this study, showing E2F score as an emerging prognostic and predictive biomarker for anti-PD-L1/PD-1 therapy and other immunotherapeutic agents (McNamara et al., 2020; Yi et al., 2018). The E2F scoring system established in this study is in harmony with immune profiles, cellular infiltration, pathway enrichment, immune treatment, and clinical prognosis.

In vitro experiments were performed to validate the results from the database. The public database analysis revealed that transcription levels of E2F1-8 in tumor tissues were higher than those in normal tissues. The multivariate analysis indicated that E2F2 and E2F8 were independent prognostic factors. The qPCR results verified the differential expression of transcription factors in tissues collected from gastric cancer patients. Cellular experiments confirmed that the transcriptional level of E2F2 and E2F8 affected cell proliferation and migration in gastric cancer cell lines. These experimental results were consistent with the database analysis. The knockdown of E2F8 promoted proliferation and migration in tumor cells, validating E2F8 as a protective prognostic factor. Previous studies and the public database analysis indicate that E2F2 also plays a protective prognostic role in gastric cancer, but E2F2-deficiency in this study weakened the proliferation and migration of gastric cancer cell lines, a result consistent with the findings of previous studies (Manicum et al., 2018; Li et al., 2021a; Wang et al., 2016). These contradictory results appear to be related to the bidirectional regulation of the cell cycle, and current evidence suggests that E2F2’s role as a tumor suppressor or tumor promoter depends on the cellular context and its interaction with other E2Fs (Azkargorta et al., 2010; Zhou et al., 2013). The E2F scoring system in this study considers the interaction of the E2F family and the TME, so it is more accurate and reliable than the prediction effect of a single transcription factor.

The E2F scoring model established in this paper reflects the immune infiltration features of individual patients, helping to determine the immune phenotypes of GC patients and guide clinical decision-making. There was a tight correlation between E2F score and the efficacy of adjuvant chemotherapy, and E2F score acted as a predictive marker for response to anti-PD-1/PD-L1 immunotherapy. E2F expression patterns were also closely linked with clinicopathological features of GC patients, such as histological subtypes, tumor grade, molecular subtypes, TMB, and survival status, which indirectly corroborated the predictive value of E2F.

Two major E2F transcription factors were chosen to validate the expression level in clinical samples via qPCR, and the results were consistent with the database, confirming the authenticity and dependability of the research. The findings of this study provide new insight into E2F expression patterns underlying TME immune infiltration and provide a new direction for the design of individualized treatments for GC patients. This study also provides a foundation for future studies on the specific mechanisms of each E2F transcription factor in TME immune response.

Conclusions

This study found that E2F transcription factors are crucial in forming TME diversity and complexity in GC, affecting prognosis and drug treatment efficacy. An E2F scoring system was also established based on E2F prognostic-related genes, which can be used to evaluate the prognosis, immune patterns, and treatment response of GC patients.

Supplemental Information

Supplemental Information 1 Patient clinical information

Click here for additional data file.

Supplemental Information 2 Primer of E2F2 and E2F8

Click here for additional data file.

Supplemental Information 3 SiRNA sequence of E2F2 and E2F8

Click here for additional data file.

Supplemental Information 4 Sequence of overexpression plasmid for E2F2 and E2F8

Click here for additional data file.

Supplemental Information 5 Kaplan–Meier survival curves of eight E2Fs

Click here for additional data file.

Supplemental Information 6 Unsupervised clustering analysis of E2F Clusters

(A–B) cumulative distribution functions (CDF) of E2F Clusters.

Click here for additional data file.

Supplemental Information 7 Unsupervised clustering analysis of gene clusters

(A) Consensus clustering matrix for k = 3. (B–C) CDF of E2F gene clusters.

Click here for additional data file.

Supplemental Information 8 Raw figure of Fig. 8

Click here for additional data file.

Supplemental Information 9 Raw data and analysis of Fig. 8

Click here for additional data file.

Supplemental Information 10 E2F2 and E2F8 plasmid DNA sequencing

Click here for additional data file.

Additional Information and Declarations

Competing Interests

Author Contributions

Human Ethics

Data Availability

The authors declare there are no competing interests.

Fanni Li conceived and designed the experiments, analyzed the data, prepared figures and/or tables, and approved the final draft.

Jun Yan analyzed the data, prepared figures and/or tables, and approved the final draft.

Jing Leng analyzed the data, prepared figures and/or tables, and approved the final draft.

Tianyu Yu performed the experiments, prepared figures and/or tables, and approved the final draft.

Huayou Zhou performed the experiments, prepared figures and/or tables, and approved the final draft.

Chang Liu performed the experiments, authored or reviewed drafts of the article, and approved the final draft.

Wenbo Huang performed the experiments, authored or reviewed drafts of the article, and approved the final draft.

Qi Sun conceived and designed the experiments, authored or reviewed drafts of the article, and approved the final draft.

Wei Zhao conceived and designed the experiments, authored or reviewed drafts of the article, and approved the final draft.

The following information was supplied relating to ethical approvals (i.e., approving body and any reference numbers):

The Ethics committee of the First Affiliated Hospital of Xi’an Jiaotong University approval to carry out the study within its facilities (Ethical Application Ref: XJTU1AF2023LSK-451).

The following information was supplied regarding data availability:

The raw data are available in the Supplementary File.

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
