# Peer review of "Expression patterns of E2Fs identify tumor microenvironment features in human gastric cancer"

_PeerJ, doi:10.7717/peerj.16911_

## Round 0.1 · original submission · Major Revisions

Please address the comments from the reviewers.

**Language Note:** PeerJ staff have identified that the English language needs to be improved. When you prepare your next revision, please either (i) have a colleague who is proficient in English and familiar with the subject matter review your manuscript, or (ii) contact a professional editing service to review your manuscript. PeerJ can provide language editing services - you can contact us at copyediting@peerj.com for pricing (be sure to provide your manuscript number and title). – PeerJ Staff

Reviewer 1 ·

Basic reporting

Manuscript written and explained the all the literature well and scientific data included well.

Experimental design

Experimental designs are as per requirements and answer the question well. Methods section describe all the information.

Validity of the findings

Overall, described and provided information are not sufficient. The findings are very preliminary and requires further experiments.

Additional comments

Manuscript id : #88415
Manuscript title: Comprehensive analysis of the expression patterns for E2Fs identifies tumor microenvironment features in human gastric cancer.
The research article submitted by Li and Yan et al. describes the expression pattern of E2Fs in tumor microenvironment in human gastric cancer (GC). Author concludes the E2F score to predict survival and treatment in GC. This study utilizes the database available from the GC patients to analyze the E2Fs expression pattern and further validate their analysis using cell lines. Overall, The Manuscript also into scope of Journal and identify the scientific questions. A set of experiments support their observation in this manuscript and data are well organized and discussed along with reference. This work should also include some important concerns to strengthen the work.
Comments
1. Author validate their analysis using the knockdown of E2F2 and E2F8 and studies their effect, if the over expression of these E2Fs reverse effect in cells? This will concrete this study further.
2. If role of either E2Fs or Transcription factor in cancers or GC studied, this information should be added in introduction section.
3. Line 210, Information does not match with the figure 1B. Data shows highest difference in E2F6 and E2F7.
4. Line 213, As per figure section, Not only E2F6 but other E2F1-E2F6, all have same mutation frequency. While E2F7 and 8 high mutation frequency. This information should also explained.

·

Basic reporting

The aim of the present article was to demonstrate the regulation patterns of 8 E2F transcription factors in gastric cancer and their association with cancer prognosis, tumor microenvironment, and treatment. Motivation for the study is well supported by the introduction and the weak point of this article is that lack of mechanism for the E2F2 and E2F8 contribution for cancer development and survival, but independently of that, suitable for publication at Peer J without further modifications.

Experimental design

In materials and methods section authors collected 12 tissue samples from the tissue bank Hospital of Xi’an Jiaotong University. It is not clear that among 12 tissue samples how many are cancer tissues and normal? Please indicate the number exactly. There is no clinical characteristics of patients. Do authors obtain the patient consent form?

Validity of the findings

Authors identified that significant upregulation of the E2F2 and E2F8 in gastric cancer tissues when compared with the normal control tissues and these findings correlated with the earlier studies and transcriptomic database. But here authors did not stated mechanism by which E2Fs (E2F2 and E2F8) contribute to development and how they influence the patient survival.

Additional comments

Nil

---

## Round 0.2 · Minor Revisions

Please address the comments of the reviewer and resubmit the manuscript with track changes along with the point by point response.

Reviewer 3 ·

Basic reporting

The paper “Expression patterns of E2Fs identify tumor microenvironment features in human gastric cancer” by Li and Yan et al. is in overall good condition and I support it for publication.

I have some minor some minor suggestions.
• In Fig. 8B. NC stands for negative control, and OE stands for overexpression; write this in the legend for easy visualization.
• Details of the plasmids used for overexpression in Fig 8, are not described anywhere in the paper, although it is mentioned that they were purchased from Company X.
• In the methodology section (2.9), list all the assays separately and provide an improved description of the reagents and instruments used.

Experimental design

NA

Validity of the findings

NA

Additional comments

NA

---

## Round 0.3 · accepted · Accept

Thank you for addressing all the reviewers' comments and resubmitting the revised manuscript.